# Religious Moderation in Indonesian Muslims

**Imam Subchi [1], Zulkifli Zulkifli [2], Rena Latifa [3,\*] and Sholikatus Sa'diyah [1]**

1   Faculty of Adab and Humanities, Universitas Islam Negeri Syarif Hidayatullah Jakarta,
    Jakarta 15412, Indonesia; imam.subchi@uinjkt.ac.id (I.S.); sholikatus.sadiyah@uinjkt.ac.id (S.S.)
2   Faculty of Social and Political Science, Universitas Islam Negeri Syarif Hidayatullah Jakarta,
    Jakarta 15412, Indonesia; zulkifli@uinjkt.ac.id
3   Faculty of Psychology, Universitas Islam Negeri Syarif Hidayatullah Jakarta, Jakarta 15412, Indonesia
\*   Correspondence: rena.latifa@uinjkt.ac.id

**Abstract:** Indonesia receives a high religious harmony index every year; however, intolerance and religious radicalism threaten this harmony. Moderate Islam (Islamic religious moderation) has become a national policy as a solution to prevent intolerance and radicalism. In this study, we aimed to determine the factors influencing religious moderation. We examined the variables of religiosity and demographics, which play essential roles in forming religious moderation. A total of 578 students at state Islamic universities in Indonesia participated in this research. We measured religiosity with the Centrality of Religiosity Scale (CRS-5) by Huber and Huber. The CRS-5 consists of five dimensions: intellect, ideology, public practice, private practice, and religious experience, which we adapted to the Indonesian language. The Religious Moderation Scale consists of three dimensions: national commitment, rejecting violence, and accommodating culture. We collected data through questionnaires that we distributed online, and we analyzed the responses using multiple regression analysis. The results show that religiosity positively affected religious moderation, meaning that religious intellectuality, ideology, public practice, private practice, and religious experience supported a person in being moderately religious and might prevent intolerance and radicalism. Socioeconomic factors (sex and parents' income) also strongly affected religious moderation.

**Keywords:** religious moderation; tolerance; intolerance; Indonesian Muslim; religiosity

## 1. Introduction

Indonesia is diverse in ethnicities, languages, religions, and beliefs. Given this condition, social harmony must be managed in Indonesia. However, intolerance, which disrupts harmony, especially between religious communities, has recently increased in Indonesia. Forms of intolerance can be more diminutive, including a lack of mutual understanding, less mutual respect, and less respect for differences in beliefs.

Several surveys conducted by the Setara Institute (2019) showed increasing intolerance among schools and students between 2007 and 2018. In addition, the number of violations of religious freedom increased from 134 cases in 2014 to 208 in 2016, and the number of intolerant actions also rose from 177 in 2014 to 270 in 2016.

One of the efforts implemented to manage and improve religious harmony was the launching of the concept and policy of religious moderation by the Indonesian Ministry of Religion. The Ministry of Religion describes the characteristics of individuals who practice religious moderation, namely if they apply religious tolerance, show national commitment, and accommodate local culture. Religious moderation aims to create a public attitude of a moderate, not extreme, understanding of religion and not glorify unlimited free-thinking in realizing a harmonious and peaceful life within the diversity in Indonesia.

Suhartawan (2021) stated that religious moderation allows people to maximize the benefits of religious and state institutions and provides for the education of future generations, the building of cross-generational collaboration, the improvement in the quality

of a more open religious understanding, and the creation of a dialogue between culture, religious and state leaders.

As a new concept and policy in Indonesia, religious moderation requires further study as research in this area is still limited. Ali (2020) measured religious moderation among 356 students at two universities in Kalimantan. As a result, the descriptive analysis provided the percentage of those considered religiously moderate. Ali found that although students had low levels of knowledge, participation, and experience in religious moderation, they had high levels of attitudes and behavior toward the dimensions of religious moderation: religious tolerance (91.5%), national commitment (95.6%), and accommodating local culture (94.9%). However, Ali did not determine the factors contributing to religious moderation.

In this study, we aimed to determine the factors influencing religious moderation. We examined the variables of religiosity and demographics that play an essential role in forming religious moderation.

Religious moderation is generally understood as the tolerance of a religious group toward different faiths while simultaneously refraining from liberalism. Thus, the concept of religious moderation, particularly among Indonesian Muslims, is defined using the ideas of the concept and the policy of religious moderation by the Indonesian Ministry of Religion. The Ministry of Religion describes the characteristics of individuals who apply religious moderation, namely if they show religious tolerance and national commitment and accommodate local culture.

The first variable that we hypothesized would predict religious moderation is religiosity. Religiosity generally refers to the observable concrete ways individuals or groups express the multiple dimensions of religion. Many definitions and measurements of the concept from Christian tradition have been developed since Glock and Stark's (1965) five dimensions of religiosity: belief, practice, knowledge, experience, and consequence. Several conceptions and measurements of Muslim religiosity have also been formulated. For instance, Riaz Hassan (2007b) explicitly drew on the work of Stark and Glock to measure Muslim religious piety. In this study, for the measurement of religiosity, we used Huber and Huber's (2012) model with five dimensions: intellectual, ideology, public practice, private practice, and religious experience, called the Centrality of Religiosity Scale (CRS). The dimensions are described as follows:

Intellectual refers to the social expectation that religious people tend to have religious knowledge and the frequency of thinking about religious topics. The ideological dimension refers to the social expectation that religious people tend to have faith in the existence and power of God and the plausibility of the presence or power of God. The public practice dimension refers to the social expectation that religious people participate in public rituals and communal activities and participate in public religious services and activities. The private practice dimension refers to the social expectation that religious people perform religious rituals and personal service styles such as prayer and meditation, including intensity and personal religious rituals. The dimension of religious experience refers to the social expectation that religious people tend to have some kind of contact with God, who emotionally influences a person's life.

Regarding the concept of religiosity by Huber and Huber (2012), we thought that those with a high level of religiosity would have a high level of religious moderation. This means that religious intellectuality, ideology, public practice, private practice, and religious experience support someone in being a moderately religious person (high in religious tolerance, national commitment, and accommodating local culture). Conversely, we believe that religious intolerance, low national commitment, and low accommodation of local culture would be prevented if a person has high religious intellectuality, ideology, private practice, and religious experience.

Golebiowska (2004) reported that religiosity is an essential predictor of tolerant behavior amongst people of faith, where tolerance is the concept of religious moderation. The study results show that religiosity positively and significantly affects religious moderation

(tolerance). Hoffman (2020) also found that religious practices, one of the dimensions of religiosity, also predict religious harmony or moderation, further supporting our hypothesis.

Because religious moderation is close to religious tolerance, several researchers have examined the relationship between religiosity and tolerance, both religious tolerance and social and political tolerance. The results of these studies on the relevance of religiosity and tolerance are complex and contradictory. The results show that religiosity can either increase or decrease tolerance. Other results reveal that religiosity is not related to tolerance (Spierings 2019). The findings may have been caused by differences in the concept of religiosity used.

Yusuf et al. (2020) found that the level of religiosity is closely related to religious intolerance. Their analysis of the effect of socio-economic characteristics and religiosity on religious intolerance found that religiosity was the only variable at the individual level that had a statistically significant and robust relationship with all types of religious intolerance. Other factors related to intolerance included income levels, quality of work, and higher education.

By contrast, Setiawan et al. (2020) found that Muslim community members supported inter-religious conflicts. Religiosity plays an essential but different role in supporting inter-religious conflicts in society, including Indonesia. Different dimensions of religiosity have different connections with religious conflicts. Although religious belief has strong influences, its effect varies. Additionally, particularistic views and intertextual fundamentalism tend to support the lawful protest. Religious centrism contributes to supporting lawful and violent protests. The religious practice of religiosity is not simply related to support of inter-religious conflict; still, a rite of the passage appears to be an essential factor contributing to inter-religious conflict rather than participation in ordinary religious rituals. Conversely, religious salience harms support for violent hostility toward outgroups.

Hoffman (2020) revealed that religious behavior in the Arab world, in general, has an ambiguous influence on tolerance. People who frequently attend mosques tend to be more intolerant of other sects than people who rarely attend. In Lebanon, the process occurs through a group identification mechanism. Attendance at mosques encourages a higher sectarian identification, making adherents feel different from other denominations. Participation in communal worship increases sectarian solidarity at the expense of sympathy for other denominations.

Conversely, private worship positively influences tolerant attitudes in the Arab world and Lebanon. "While congregational prayer tends to enhance sectarian identity and reinforce intolerance, citizens who obey individually—at least, those who pray privately—tend to be more tolerant of other sects than their less obedient counterparts." In a sectarian environment, private religious practice substantially positively affects tolerance.

In research on various aspects of the impact of religiosity on social tolerance in the Middle East and North Africa, Spierings (2019) found that the degree of religious identification does not harm social tolerance except for a few cases where Islamist power is present. Under normal circumstances, those with religious beliefs tend to be more tolerant of others but less tolerant if they feel threatened or repressed. Attendance at mosques harms ethnic-religious social tolerance and even strengthens it if a conservative Islamist state coercively regulates religion and its content, such as sermons.

Hassan (2007a) studied the level of religiosity of Muslim communities in various parts of the world and its relation to modernity, Muslim umma awareness, and intolerance (blasphemy). Hassan found that religiosity does not affect modernity, especially the Human Development Index. However, as many scholars affirmed, he found that religiosity positively affects intolerance, especially blasphemy. PPIM research has produced different results: students with a high religiosity tend to have moderate religious attitudes and behavior. Thus, religiosity has a positive effect on religious tolerance. Due to the contradictions in the previous findings on the impact of religious tolerance, we examined the effects of the religiosity variable on religious moderation in this study. We also considered several demographic variables as factors influencing religious moderation. The

first demographic variable was sex (identified as male and female). Theories show that sex substantially affects religious moderation. According to Çavdar (2010), women show higher religious moderation than men. Sex influences one's level of religious moderation. Van der Jagt et al. (2018) also reported a notable effect of sex on religious moderation, a concept of religious harmony/disharmony in that study.

The second demographic variable we hypothesized as affecting religious moderation was student organizations. We expected religious moderation to be influenced by sociology, and a student organization is one of the social constructs. We theorized that student organizations positively and significantly affect religious moderation, meaning that students involved in student organizations have higher religious moderation than those who are not. This hypothesis is supported by Hendrastomo et al. (2019), who showed that the more social the individual, the more moderate they are through association with a student organization.

The third and fourth demographic factors that we theorized would predict religious moderation were the economic status variables, namely parental income and the amount of allowance, respectively. We hypothesized that parental income and allowance amount would positively and significantly predict religious moderation. The higher the parental income and the higher the allowance, the higher the religious moderation. Based on the annual survey of religious harmony by the Ministry of Religious Affairs (2020), socio-economic factors affect the religious harmony index so that harmony can describe the religious moderation index.

## 2. Materials and Method

### 2.1. Participants and Sampling Procedures

We employed a quantitative approach by administering a survey in July 2020 to all students of Indonesia's state Islamic higher educational institutions. The sample was sixth-semester undergraduate Muslim students at state Islamic higher educational institutions in the western, middle, and eastern areas of Indonesia, totaling 578 students (N = 578). We used a multistage systematic sampling technique. We started by selecting state Islamic universities or institutes in western, middle, and eastern Indonesia: UIN Arraniry Banda Aceh, UIN Raden Fatah Palembang, and UIN Syarif Hidayatullah Jakarta in western Indonesia, UIN Alauddin Makasar and UIN Mataram in middle Indonesia, and IAIN Ambon and IAIN Sorong in eastern Indonesia. We collected data through questionnaires consisting of closed-ended questions distributed to students through Google Forms.

### 2.2. Measurement and Data Analysis

We processed and analyzed the collected data using multiple regression analysis to examine the effect of several independent variables, namely religiosity and socio-economic factors (sex, student organization, parents' income, and allowance amount), on the dependent variable, religious moderation. The hypothesis we examined was that religiosity, sex, participation in a student organization, parents' income, and allowance amount affect religious moderation. We formulated this hypothesis as follows:

$$Y = a + b_1X_1 + b_2X_2 + b_3X_3 + b_4X_4 + b_5X_5 + e$$

After we collected and processed the data, we tested the validity of the items, especially regarding religiosity and religious moderation, using confirmatory factor analysis (CFA). We used MPlus 8.4 software to perform the CFA and SPSS v 25 to test the hypothesis.

Unlike the items of religiosity, which were all valid, not all religious moderation items were valid. Below, we describe the items we used to measure religiosity that we modified from Huber and Huber (2012).

#### 2.2.1. Religiosity Construct

This study measured religiosity with the Centrality of Religiosity Scale (CRS-5) (Huber and Huber 2012). Huber and Huber (2012) provided alternatives of 15, 10, or 5 items to

measure religiosity. The CRS-5 has five items and consists of 5 dimensions—intellectual, ideology, public practice, private practice, and religious experience—which we adapted to the Indonesian language.

For the translation procedure, we used first-round back-translation (Appendix A). For the responses, a Likert scale with four points was used (1 = strongly disagree, 2 = disagree, 3 = agree, and 4 = strongly agree).

To test the construct validity of religiosity, we used a multiple-factor model of analysis. Using CFA, we found the model fit criteria were: chi-square > 0.05, $p$-value < 0.01, RMSEA < 0.05, and CFI and TLI > 0.90.

### 2.2.2. Religious Moderation

The religious moderation scale was constructed from the Ministry of Religious Affairs (2019) definition of religious moderation. This scale consists of 3 dimensions: national commitment, nonviolence, and accommodating local culture. This study tested whether the fifteen items are truly unidimensional, meaning that they only measure religious moderation. The responses to the questions were provided using a Likert scale with four points (1 = strongly disagree, 2 = disagree, 3 = agree, and 4 = strongly agree).

To test the construct validity of religious moderation scale, we used a multiple-factor model of analysis. Using CFA, we found the model fit criteria were: chi-square > 0.05, $p$-value < 0.01, RMSEA < 0.05, and CFI and TLI > 0.90.

### 2.2.3. Sex

Sex was one of the demographic variables we tested in the regression model in this study. We used a dummy coding of 1 = man and 2 = woman to code sex as a dichotomous categorical variable.

## 3. Results

### 3.1. Scale Validation

#### 3.1.1. Central Religiosity Scale Validation Result

From the results of the CFA with the one-factor model, we found that the model did not fit, with chi-square = 173.46, df = 5, $p$-value = 0.00000, and RMSEA = 0.242. Therefore, we modified the model so the measurement errors on some items were free to correlate with each other. As such, we obtained a fit model with chi-square = 0.90, df = 2, $p$-value = 0.63679, and RMSEA = 0.000. The next step was to determine the validity of each item of religiosity. The results are provided in Table 1.

**Table 1.** Validity of religiosity items.

| Item | Coefficient | SE | t-Value | Result |
|:---:|:---:|:---:|:---:|:---:|
| 1 | 0.80 | 0.05 | 17.57 | Valid |
| 2 | 0.57 | 0.04 | 13.26 | Valid |
| 3 | 0.72 | 0.04 | 16.52 | Valid |
| 4 | 0.70 | 0.05 | 15.06 | Valid |
| 5 | 0.62 | 0.05 | 13.67 | Valid |

Note: valid = t-value > 1.96.

Based on the table, we found that all five items had a t-score higher than 1.96 and were all valid religiosity measures.

#### 3.1.2. Religious Moderation Scale Validation Result

From the CFA results with the one-factor model, we found a model that did not fit, with chi-square = 1222.64, df = 90, $p$-value = 0.00000, and RMSEA = 0.148. Therefore, we modified the model, where the measurement errors on some items were free to correlate with each other. This produced a model that fit with chi-square = 1.22, df = 18, $p$-value = 1.00000, and RMSEA = 0.000. We report the validity test of each item of religious moderation in Table 2.

**Table 2.** Validity of religious moderation items.

| Item | Coefficient | SE | t-Value | Result |
|:---:|:---:|:---:|:---:|:---:|
| 1 | 0.65 | 0.05 | 12.39 | Valid |
| 2 | 0.22 | 0.04 | 5.04 | Valid |
| 3 | −0.13 | 0.04 | −2.91 | Invalid |
| 4 | 0.45 | 0.04 | 10.80 | Valid |
| 5 | 0.72 | 0.04 | 16.79 | Valid |
| 6 | 0.60 | 0.08 | 7.42 | Valid |
| 7 | 0.36 | 0.05 | 7.36 | Valid |
| 8 | 0.33 | 0.05 | 6.99 | Valid |
| 9 | −0.14 | 0.05 | −2.93 | Invalid |
| 10 | 0.87 | 0.04 | 22.91 | Valid |
| 11 | 0.61 | 0.04 | 15.15 | Valid |
| 12 | 0.57 | 0.05 | 11.25 | Valid |
| 13 | 0.77 | 0.04 | 19.11 | Valid |
| 14 | 0.40 | 0.04 | 9.00 | Valid |
| 15 | 0.76 | 0.04 | 19.39 | Valid |

Note: valid = t-value > 1.96.

Based on the results in Table 2, we found two invalid items, 3 and 9, because these items received a t-value score of less than 1.96. The other items were valid, meaning they could measure the concept or variable of religious moderation in this study. Thus, we excluded items 3 and 9 from the measurement of religious moderation.

### 3.2. Test of Hypothesis

Initially, we discovered that most students at state Islamic higher education institutions could be moderate (Table 3). In detail, we found 108 respondents (18.7%) in the low category, 331 respondents (57.3%) in the medium category, and 139 respondents (24%) in the high-religiosity category. Similarly, with nearly the same numbers and percentages, the level of religious moderation tended to be moderate: 110 respondents (19%) were categorized as low, 347 respondents (60%) as moderate, and 121 respondents (20.9%) as high.

**Table 3.** Religiosity and religious moderation.

| Variable | Frequency % | | |
|:---:|:---:|:---:|:---:|
| | **Low** | **Moderate** | **High** |
| Religiosity | 108 (18.7%) | 331 (57.3%) | 139 (24%) |
| Religious Moderation | 110 (19%) | 347 (60%) | 121 (20.9%) |

To test the research hypothesis, we used multiple linear regression analysis. In regression analysis, three aspects are considered. First, the $R^2$ determines the variation in the dependent variable caused by variation in the independent variables. Second, the results of the F-test are obtained, which determine whether the $R^2$ is significant. Third, the *T*-test is performed to determine the regression coefficient of each independent and its significance.

We tested the hypothesis by obtaining the $R^2$, which showed the percentage of variation in the dependent variable in this study, religious moderation, caused by all independent variables (socio-economic characteristics and religiosity). The $R^2$ is provided in Table 4.

**Table 4.** $R^2$ results in this study.

| Model | R | $R^2$ | Adjusted $R^2$ | SE of the Estimate |
|:---:|:---:|:---:|:---:|:---:|
| 1 | 0.294 [a] | 0.086 | 0.078 | 8.84730 |

[a] Dependent variable: religious moderation.

Based on the results in Table 4, we found that the $R^2$ was 0.086 or 8.6%. This means that 8.6% of the variation in religious moderation could be explained by sex, student

organization, parental income, allowance amount, and religiosity. The remaining 91.4% was explained by other variables not considered in this study. After obtaining the $R^2$, we conducted the F test to analyze the impact of all the studied independent variables on the dependent variable, religious moderation. The results of the F test are provided in Table 5.

**Table 5.** ANOVA results.

| Model | Sum of Squares | df | Mean Square | F | Sig. |
|---|---|---|---|---|---|
| Regression | 4229.992 | 5 | 845.998 | 10.808 | 0.000 [b] |
| Residual | 44,773.151 | 572 | 78.275 | | |
| Total | 49,003.143 | 577 | | | |

[b] Predictors: (constant), sex, student organization, parents' income, amount of allowance, and religiosity.

Based on the $R^2$ and its significance from multiple regression analysis, we considered the regression coefficient of each independent variable. The regression coefficients for each independent variable obtained from the *T*-test are shown in Table 6.

**Table 6.** Regression coefficients for each independent variable.

| Model | Unstandardized Coefficients | | Standardized Coefficients | | |
|---|---|---|---|---|---|
| | B | SE | Beta | T | Sig. |
| Constant | 36.401 | 2.804 | | 12.984 | 0.000 |
| Sex | −0.2145 | −0.749 | −0.116 | −2.865 | 0.004 * |
| Student Organization | 0.029 | 0.140 | 0.008 | 0.207 | 0.836 |
| Parental Income | 2.034 | 0.899 | −0.096 | 2.263 | 0.024 * |
| Amount of Allowance | 0.301 | 0.649 | −0.020 | 0.464 | 0.643 |
| Religiosity | 0.282 | 0.044 | 0.255 | 6.358 | 0.000 * |

Dependent variable: religious moderation. *, significant (sig. < 0.05).

Based on Table 6, we determined the regression coefficient for each independent variable, and we formulated the regression equation as follows:

Religious moderation = 36.401 − 0.2145 sex + 0.029 student organizations + 2.034 parents' income + 0.301 amount of allowance + 0.282 religiosity.

We provide the significant regression coefficient in the significant value column in Table 6, where a value < 0.05 means that the coefficient had a significant effect on the dependent variable. In the following, we explain the regression coefficient for each independent variable:

1. The regression coefficient for sex was 0.004. Thus, the null hypothesis (Ho), which stated that sex had no significant effect on religious moderation, was rejected. This means that sex has a significant effect on religious moderation.
2. The regression coefficient of student organization was 0.836. Thus, the null hypothesis (Ho), which stated that student organizations had no significant effect on religious moderation, was accepted.
3. The regression coefficient of the parents' income was 0.024. Thus, the null hypothesis (Ho), which stated that parental income had no significant effect on religious moderation, was rejected. This means that the parent's income significantly affected religious moderation. The direction of the positive coefficient showed that the higher the parental income, the higher the level of students' religious moderation, and vice versa.
4. The regression coefficient of the amount of allowance was 0.643. Thus, the null hypothesis (Ho), which stated that the amount of allowance had no significant effect on religious moderation, was accepted.
5. The regression coefficient for religiosity was 0.000. Thus, the null hypothesis (Ho), which stated that religiosity had no significant effect on religious moderation, was rejected. This means that religiosity had a significant effect on religious moderation.

The direction of the positive coefficient showed that the higher the level of students' religiosity, the higher the level of their religious moderation, and vice versa.

Based on Table 6 and the explanation above, we found that three variables had a significant effect on religious moderation: religiosity, sex, and parental income; the other two variables, student organization and the allowance amount, had no significant effect on religious moderation among students at state Islamic higher educational institutions in Indonesia.

## 4. Discussion

The research results show that religious moderation can be partly explained by religiosity and socio-economic factors.

First, we found that religiosity positively affected religious moderation. Those who had a high level of religiosity tended to have a high level of religious moderation. This means that religious intellectuality, ideology, public practice, private practice, and religious experience support a person in being moderately religious (high in religious tolerance, national commitment, and accommodating local culture).

We found that every aspect of religiosity explained by Huber and Huber (2012), namely intellectual, ideology, public practice, private practice, and religious experience, correlated with a religiously moderate person. Religious intolerance, low national commitment, and low accommodating local culture could be prevented if a person shows high religious intellectuality, ideology, private practice, and religious experience. Furthermore, researchers can test what dimension of religiosity is related to extremist behavior in the future.

Our findings are close to those of PPIM, showing that students with a high religiosity tend to have moderate religious attitudes and actions (PPIM 2018). Other researchers explained religiosity as the essential variable that supports religious intolerance at the individual level. Yusuf et al. (2020) found that the level of religiosity was closely related to religious intolerance. Their analysis of the effect of socio-economic characteristics on religious intolerance showed that religiosity was the only variable at the individual level that had a statistically significant and robust relationship with all types of religious intolerance.

Setiawan et al. (2020) found that members of the Muslim community tended to support inter-religious conflict. Religiosity plays an important but varying role in supporting inter-religious conflicts in society, including Indonesia. Other dimensions of religiosity have different relationships with religious conflict. Although religious beliefs significantly influence the support of religious conflict, their influence varies. Whereas some views and fundamentalism favor legitimate protest, religious centrism contributes to supporting the legitimate and violent protest. A rite of passage is a contributing factor to inter-religious conflict.

Religious practices support inter-religious conflicts, whereas religious salience reduces the support for violent hostility toward outside groups. The differences in the findings of these studies are due to differences in the concept of religiosity used. We used Huber and Huber's (2012) concept of religiosity in this study.

Furthermore, Hoffman (2020) revealed that religious behavior in the Arab world, in general, has an ambiguous influence on tolerance. People who frequent a mosque tend to be more intolerant of other sects than those who rarely attend. In Lebanon, this process occurs through a group identification mechanism. Attendance at mosques encourages higher levels of sectarian identification, making them feel different from other denominations. Participation in communal worship increases sectarian solidarity at the expense of sympathy for other denominations.

On the other hand, private worship positively influences tolerance in the Arab world and Lebanon. It is much similar to the Indonesian context. Still, our findings confirm that one must clearly understand his religion (as described in the intellectual and ideological dimensions), have good religious practices (public and private), and experience to be more moderate in thinking and behaving. Further research is a recommendation to gain a complete understanding of these dynamics.

This study found that religiosity was not the only variable supporting religious moderation: sex also had a significant effect on religious moderation. The beta score showed that men are more moderate than women (coding 1 for men and 2 for women). This finding somewhat differs from Huber and Krech (2009), based on population-representative data from 21 countries, including Indonesia. Their findings show that women were slightly more pluralistic than men on the attitude level (stand. Beta +0.04), and men were somewhat more fundamentalist than women (stand. Beta −0.03). This difference should be further discussed and investigated. We suspect that men have more access to and participation in public in the Indonesian Muslim context, especially regarding religion and state participation. This may lead to men being more moderate, especially if we consider the dimension of religious moderation that we used in this research: national commitment, rejecting violence, and accommodating local culture.

In this study, parental income had a significant effect on religious moderation. The beta value was negative, meaning that when parental income increases will reduce religious moderation. Ideally, parents' income would allow their children to receive high-quality religious education and interact with people of various religions. However, in Indonesia today, many people in the upper economic circles prefer religious/spiritual conversion, becoming fundamentalists. They send their children to inclusive schools that provide fundamentalist-based religious teachings.

The negative influence of parental income is supported by the annual survey of religious harmony by the Agency for Religious Research and Development (2019), which found that socio-economic factors affect the religious harmony index. However, these findings do not further explain which socio-economic factors influence or direction the index.

Finally, we found that student organizations and the allowance received from parents were not related to religious moderation. A student's involvement in an organization did not influence the student's religious status (e.g., moderate/extreme). We suggest that future studies question the value of the student organization and not just the students' participation in an organization. The number of organizations in which students participate may also be valuable information. Hendrastomo et al. (2019) reported that the more social the individual, in this case, through the association with student organizations, the more moderate they are. This might be related to the number of organizations in which students participate.

The allowance amount could not explain religious moderation. This means that one may become a religious moderate regardless of their allowance. Conversely, extremists may have either a rich or poor background in the Indonesian context.

## 5. Conclusions and Outlook

Our findings show that religiosity has a significant positive effect on religious moderation. This finding notably differs from those obtained in most previous studies that upheld an adverse effect of religiosity, especially the dimensions of congregational prayer and rite of passage, on religious tolerance, both external and internal. This finding also emphasizes the Centrality of Religiosity Scale by Huber and Huber (2012), supporting the existence of harmony, tolerance, and radicalism prevention. Individuals should clearly understand their religion (as explained by the intellectual and ideology dimensions), perform religious practices (public and private), and engage in experiences to become more moderate in their thinking and behavior. Further research is recommended to gain a complete understanding of this dynamic.

**Author Contributions:** Conceptualization, I.S. and Z.Z.; Data curation, Z.Z., R.L. and S.S.; Formal analysis, Z.Z. and R.L.; Funding acquisition, Z.Z. and S.S.; Investigation, R.L. and S.S.; Methodology, R.L.; Project administration, S.S.; Software, R.L.; Supervision, I.S. and Z.Z.; Validation, R.L.; Writing—original draft, I.S. and Z.Z.; Writing—review & editing, Z.Z. and R.L. All authors have read and agreed to the published version of the manuscript.

**Funding:** This research was funded by UIN Syarif Hidayatullah Jakarta.

**Institutional Review Board Statement:** The study was conducted in accordance with the Declaration of Helsinki and approved by the Institutional Review Board statement (PVC Research and Community Engagement, UIN Syarif Hidayatullah Jakarta; Ethical Clearance No. B-381/LP2M/HM-01.5/12/2021).

**Informed Consent Statement:** Informed consent was obtained from all subjects involved in the study. We anticipated no risks associated with the study. The information we collected in this study was anonymous, and the data collected were maintained and kept for research or educational purposes only.

**Data Availability Statement:** The data presented in this study are available upon request from the corresponding author.

**Acknowledgments:** We acknowledge the support given by UIN Syarif Hidayatullah Jakarta for the 2020 research grant.

**Conflicts of Interest:** The authors declare no conflict of interest.

## Appendix A

**Table A1.** The Centrality of Religiosity Scale (CRS).

| No | Dimension | Original Item | Modified Item | Indonesian Translation |
|---|---|---|---|---|
| 1 | Intellectual | How often do you think about religious issues? | I often think/study religious topics. | Saya sering berpikir/mempelajari tentang topik-topik keagamaan. |
| 2 | Ideology | To what extent do you believe that God exists? | I believe in the proof of God's power. | Saya percaya pada bukti-bukti kekuasaan Tuhan. |
| 3 | Public practice | How often do you take part in public services? | I often participate in religious activities in the community. | Saya sering berpartisipasi dalam aktivitas keagamaan di masyarakat. |
| 4 | Private practice | How often do you pray? | I often perform obligatorily and recommended prayers. | Saya sering melaksanakan shalat wajib dan shalat sunnah. |
| 5 | Experience | How often do you experience situations in which you have the feeling that God intervenes in your life? | I often experience situations where I feel God's intervention in my life | Saya sering mengalami situasi dimana saya merasakan ada campur tangan Tuhan dalam kehidupan saya. |

**Table A2.** Religious moderation scale.

| No | Dimension | Indonesian Versions | English Versions |
|---|---|---|---|
| 1 | National Commitment | 1. Saya menerima Pancasila sebagai ideologi negara. | 1. I accept Pancasila as the state ideology |
| | | 2. Saya menghargai akan keragaman ideologi lain yang ada di negara ini. | 2. I appreciate the diversity of other ideologies that exist in this country. |
| | | 3. Ideologi lain selain Pancasila adalah salah (reverse item). | 3. Other ideologies except Pancasila are wrong (reverse item). |
| 2 | Rejecting violence | 4. Membela agama dengan cara kekerasan adalah cara yang salah. | 4. Defending religion by violence is wrong. |
| | | 5. Agama tidak mengajarkan saya kekerasan, maka saya bersikap bijak dalam bertindak. | 5. Religion does not teach me violence, so act wisely. |
| | | 6. Saya memandang bahwa agama lain adalah musuh bagi agama saya, maka saya harus perangi (reverse item), | 6. I think that other religions are enemies of my religion, so I have to fight them (reverse item). |
| | | 7. Saya rela melakukan tindak kekerasan atas dasar agama saya (reverse item). | 7. I am willing to commit acts of violence based on my religion (reverse item). |
| | | 8. Tindak kekerasan atas dasar agama tidak dibenarkan. | 8. Acts of violence based on religion are not justified. |
| | | 9. Apabila agama saya dihina, maka saya siap membela agama saya dengan cara apapun (reverse item). | 9. If my religion is insulted, then I am ready to defend my religion in any way (reverse item). |

**Table A2.** *Cont.*

| No | Dimension | Indonesian Versions | English Versions |
|----|-----------|---------------------|------------------|
| 3 | Accommodating Local Culture | 10. Agama mengajarkan saya untuk menghargai setiap budaya lokal. | 10. Religion teaches me to respect every local culture. |
| | | 11. Kebudayaan lokal tidak dibenarkan dalam agama saya (reverse item). | 11. Local culture is not allowed in my religion (reverse item). |
| | | 12. Setiap budaya memiliki nilai yang baik. | 12. Every culture has good values. |
| | | 13. Saya menghargai keragaman budaya lokal yang ada. | 13. I appreciate the diversity of local cultures. |
| | | 14. Apabila kebudayaan lokal bertentangan dari agama saya, maka saya tetap menghargai budaya tersebut. | 14. If the local culture conflicts with my religion, I still respect that culture. |
| | | 15. Bersikap toleransi dengan banyak kebudayaan adalah tindakan yang tepat. | 15. Being tolerant of many cultures is the right thing to do. |

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
