# Peer review of "Religious Moderation in Indonesian Muslims"

_religions, doi:10.3390/rel13050451_

Round 1

Reviewer 1 Report

I commend the authors for their interesting contribution to Religious Moderation in Indonesian Muslims. This is extensive research and will be helpful in the academe.

Here are some comments and suggestions which can help improve the paper.

  1. The context of the study has been clearly stated in the introduction. However, the research objectives are not clear. Can you align the purpose written in the abstract with the paper? The paper must clearly identify the goals, theoretical framework, and hypothesis on pages 2-3.
  2. The main focus of the paper is religious moderation. However, the methodology did not elaborate on the material used to measure religious moderation. Unless the authors developed this scale and needed to be unidentified, this part can be added once revised.
  3. In relation to the number 1 comment --- In the abstract and theoretical framework, it is noted that the paper seeks to determine the influencing factors of religious moderation. The demographic variables include gender, student organization, and economic status (parent's income and amount of allowance). However, only two influencing factors (gender and parent's income) have been included in the discussion and conclusion. Kindly insert a short discussion about the two remaining influencing factors, and this will complete the picture of your study. 

Author Response

- "The paper must clearly identify the goals, theoretical framework, and hypothesis on pages 2-3”. We underline the goals, theoretical framework and hypothesis on pages 2-3. We also add more explanation about religiosity, tolerance and religious moderation on pages 3-5.   The paper aims to determine the influencing factors of religious moderation. Researchers examine the variables of religiosity and demographics variables that have an important role in the formation of religious moderation. (page 2)   We use a theoretical framework by Golebiowska (2004) which reported that religiosity is an important predictor of tolerant behavior amongst people of faith, in which case tolerance is the concept of religious moderation. The result of the study shows that religiosity positively and significantly affects religious moderation (tolerance). Hoffman (2020) also found that religious practices, which is one of the dimensions of religiosity, also predict the religious harmony or moderation which further support this research’s hypothesis. (Page 3)   We hypothesize that religiosity positively and significantly affects religious moderation, meaning that the more religious an individual is, the more moderate in religion he will be (not excessively highlighting his religious behavior or not being extreme). (page 3)   - "Religious moderation. However, the methodology did not elaborate on the material used to measure religious moderation. Unless the authors developed this scale and needed to be unidentified, this part can be added once revised”. We underline the additional explanation on page 6. The religious moderation scale constructed from the definition of religious moderation by Indonesian Ministry of Religious Affairs (2019). This scale consists of 3 dimensions, namely national commitment, non-violence and accommodating to local culture. In this study, we tested whether the fifteen items are truly unidimensional, meaning that it only measures religious moderation.   - The demographic variables include gender, student organisation, and economic status (parent's income and amount of allowance). However, only two influencing factors (gender and parent's income) have been included in the discussion and conclusion. Kindly insert a short discussion about the two remaining influencing factors, and this will complete the picture of your study."
We underline the additional explanation on page 11. This study found that student organization and the amount of allowance from parents are not related to religious moderation.  First, we explained about the involvement of a student in an organization, which can not prevent the student from becoming a moderate or non-moderate. We suggest for future research that the value of the student organization needs to be questioned rather than only questions about student’s participation in an organization. The number of the organizations that students participate in is also interesting to know. Since Hendrastomo (2019) research shows that the more social the individuals are, in this case it is shown through the association with student organizations, the more moderate they are. This might be related to the number of the organizations that students participated in. Second, regarding to the amount of allowance also cannot explain the religious moderation. Meaning that ones may become religious moderate not based on the money that they have. Vice versa, in Indonesian context, extremists can also come from rich or poor backgrounds.

Reviewer 2 Report

I am very glad to have had the chance to read this and enjoyed the article. This article needs clarification in a number of places (I have outlined these below). I encourage the authors to continue to work with this material.

The biggest problem is the lack of an attempt to define concepts like moderation, intolerance in the specific terms of Indonesian realities. Because there is no connection to real situations, the article comes across as a long discussion of numbers. The reader is not able toknow what is meant by intolerance and moderation.

The article is sloppy in places. ‘The index of religious harmony in Indonesia is relatively high, namely 72.20 in 2017, 70.90 in 2018, and 73.93 in 2019’…What does this mean? The reader is not given any chance to interpret this set of numbers.

Findings ought to be restricted in relevance to students of the institution…for example, the religiosity question “how often do you think about religion”…I mean all these people probably think about it every day as they are students at Islamic universities. Furthermore, these campuses are known to be sites of moderation in the spectrum of Indonesian universities. Other universities will give different findings.

Gotong royong is not translated or explained.

About the scales: Acceptance of the state ideology is equated with moderation…this cannot make sense…it equates political ideology with religious moderation…the logic is wrong…mistakes religion for political dispositions.

Author Response

- Explain the concepts of "moderation and intolerance in the specific terms of Indonesian realities”. We underline the additional explanation on page 1-2.
Indonesian reality is the existence of the diversity of ethnic groups, ethnicities, languages, religions and beliefs. However, lately, it appears that there is an increase in intolerance that can disrupt the harmony of life between religious communities in Indonesia. The form of intolerance can be as less-mutual understanding, less in mutual respect, and less-respect for differences in beliefs (regarding to all the diversity mentioned). —> 1st paragraph of page 1.
The example of intolerance in Indonesia: violations of religious freedom, the rise of extreme ideologies, conservative and exclusivity among college students, intolerance attitude toward minority muslim groups such as Shia and Ahmadiyah. (Page 1-2)     - Explanation of the statement "The index of religious harmony in Indonesia is relatively high”.  We underline the additional explanation on page 2.
The Religious Harmony Index is a national survey that measures the level of religious harmony in Indonesia. The three dimensions of the Index include tolerance, equality, and mutual cooperation (gotong royong). The tolerance aspect is the lowest between other dimensions (the equality and mutual cooperation found high scores in 2020 research).     - Translation or explanation of "Gotong royong”. We underline the additional explanation on page 2.
The gotong royong is one of Indonesian values, defined as mutual cooperation/mutual assistance, working hand in hand. 

Reviewer 3 Report

Overall, the article appears to accomplish what it sets out to do--namely, to show that religiosity is positively correlated with religious moderation.

I am not a quantitative researcher, so I cannot say anything about the statistical parts of the research design.  Assuming someone with expertise in that area finds this acceptable, then I would say there is enough of interest here to publish the piece.

I would recommend perhaps a bit more qualitative/narrative context, specifically in two areas.  As a scholar of Indonesia's religious history and culture, I understand why the author chose the indicators of religious moderation that the author did; but explaining why these indicators "count" as markers of moderation in the Indonesian context would be helpful.  Second, a bit more explanation of the findings would be helpful.  Why, for example, does the author think that the data presented here differ from findings in other studies in which religiosity (or certain kinds of religiosity--communal versus private, for example) correlates with decreased moderation.  Some of this is implied, but it would be good to conclude the discussion with a more comprehensive discussion of the findings in terms of the narrative interpretation of the author.

Finally, a few specific points:

-On p. 3 the author writes that "It is commonly agreed," but just before that the author points out that the literature doesn't actually agree on the point in question--i.e., the relationship between religiosity and religious moderation--and at the end of the paper the author suggests that the literature generally agrees that high religiosity translates into low moderation, and thus offers this study as a foil.  It would be good to smooth all of this out.  What does the literature generally say, and how is that similar or different to this study?

-On p. 5, gender is not "the biological characteristics" of different people.  By definition, gender is how biological differences are read culturally.  Maybe the author actually means "sex" here?  Or maybe the author means "gender" and should be more careful in how the author defines the term.

-Throughout the article, there are sections marked in red ink.  Is that on purpose?  If so, can the author explain why?  If not, please fix this.

Author Response

- “I would recommend perhaps a bit more qualitative/narrative context, specifically in two areas.  As a scholar of Indonesia's religious history and culture, I understand why the author chose the indicators of religious moderation that the author did; but explaining why these indicators "count" as markers of moderation in the Indonesian context would be helpful.   We underline the additional explanation on page 2. Religious moderation aims to bring the public to a moderate understanding of religion, not extreme in attitude, and also not glorifying unlimited free thinking in the context of realizing a harmonious and peaceful life in diversity in Indonesia. Suhartawan (2021) stated that religious moderation directs people to maximize the benefits of religious and state institutions; educate future generations; build cross-generational collaboration; improve and improve the quality of a more open religious understanding and also create a culture of dialogue between religious leaders and state leaders.   Second, a bit more explanation of the findings would be helpful.  Why, for example, does the author think that the data presented here differ from findings in other studies in which religiosity (or certain kinds of religiosity--communal versus private, for example) correlates with decreased moderation.  Some of this is implied, but it would be good to conclude the discussion with a more comprehensive discussion of the findings in terms of the narrative interpretation of the author.”
  We re-write the discussion in an underlined paragraph on page 9-10. Hopefully it will be more clear and comprehensive. Thank you for the feedback, as the first draft is a very technical and superficial explanation.     - “On p. 3 the author writes that "It is commonly agreed," but just before that the author points out that the literature doesn't actually agree on the point in question--i.e., the relationship between religiosity and religious moderation--and at the end of the paper the author suggests that the literature generally agrees that high religiosity translates into low moderation, and thus offers this study as a foil.  It would be good to smooth all of this out.  What does the literature generally say, and how is that similar or different to this study?”   Thank you for the feedback, as the first draft is not really easy to understand, we rewrite the flow of the introduction and delete the paragraph that the reviewer mentioned.  Please find our revised version of page 2-3. We explain what literature generally says and how it is that similar or different to this study. Thank you.  
- “On p. 5, gender is not "the biological characteristics" of different people.  By definition, gender is how biological differences are read culturally.  Maybe the author actually means "sex" here?  Or maybe the author means "gender" and should be more careful in how the author defines the term.   The authors delete the definition, as it not really important to further explanation in this case. Author only add expalanation that in this research author identified gender as male and female (as commonly in Indonesian culture). We underline the revision on page 3.

Round 2

Reviewer 2 Report

The research is not well framed. For example, we are told the research is about religious moderation, then a page or two later, we are told that the research is about the Ministry of Religion's definition of moderation...a wholly different thing...we can talk about institutional priorities in the state context....but these reflections are missing. I also am not convinved by the quantitative method.

Author Response

Dear Reviewer,   Thank you very much for your constructive and valuable feedback.   We are sending you the revised manuscript of our article. We have tried our best to follow your suggestions. We also done the English editing by MDPI. Hopefully, the revised manuscript meets your expectations.   Thank you for your kind consideration.   Best Regards,

Author(s)
